rsos.royalsocietypublishing.org

palaeontology

Tetrapodomorpha, Carboniferous, Aïstopoda, Lepospondyli, Joggins, Nova Scotia

**Author for correspondence:**
Jason D. Pardo
e-mail: jdpardo@ucalgary.ca

# A basal aïstopod from the earliest Pennsylvanian of Canada, and the antiquity of the first limbless tetrapod lineage

## Jason D. Pardo[1,2] and Arjan Mann[3]

[1]Department of Comparative Biology and Experimental Medicine, and [2]McCaig Institute of Bone and Joint Health, University of Calgary, Calgary, Canada
[3]Department of Earth Sciences, Carleton University, 1125 Colonial By Drive, Ottawa, Canada

 JDP, 0000-0002-2665-8893; AM, 0000-0002-3511-0635

Earliest Pennsylvanian (Bashkirian) vertebrate fossil assemblages of the Joggins Formation (Cumberland Group) of Nova Scotia, Canada, have long been noted for the unique representation of the earliest known crown amniotes, but the overall vertebrate fauna remains poorly understood. The vast majority of Joggins vertebrates have historically been assigned to the Microsauria, a group originally established by Dawson specifically to accommodate the abundant, diminutive fossils of the Joggins Formation. As the Microsauria concept has evolved, some Joggins taxa (e.g. the eureptile *Hylonomus lyelli*) have been removed from the group, but many of the Joggins 'microsaurs' remain unrevised, obscuring the true diversity of the earliest Pennsylvanian tetrapod fauna. Here we amend part of this problem by revisiting the morphology of Dawson's 'microsaur' *Hylerpeton longidentatum*. This taxon, represented by the anterior half of a left hemimandible, is here reinterpreted as a plesiomorphic aïstopod and assigned to a new genus, *Andersonerpeton*. *A. longidentatum* shows a surprisingly primitive anatomy of the lower jaw, retaining a parasymphyseal fang pair on the dentary, an adsymphyseal bone bearing a denticle field, fangs on all coronoids and parasymphyseal foramina, as well as a prearticular which extends far anterior along the coronoid series. However, several aïstopod characters can also be seen, including a lack of sculpturing on the dentary and a reduced number of recurved, weakly socketed teeth. The anatomy of *A. longidentatum* corroborates recent phylogenetic work which has placed the origin of aïstopods within the Devonian fin-to-limb transition but preserves a mosaic of characteristics suggesting an even earlier divergence. The presence of an aïstopod in the Joggins

rsos.royalsocietypublishing.org

R. Soc. open sci. 5: 181056

fauna expands the taxonomic diversity of the Joggins fauna and suggests that Joggins may preserve a more typical Carboniferous fauna than previously thought.

# 1. Introduction

The transition between the Lower Carboniferous (Mississippian) and Upper Carboniferous (Pennsylvanian) marks the transition from faunas dominated by archaic tetrapod groups spanning the assembly of the tetrapod body plan to faunas dominated by representatives of the tetrapod crown group which preserve the origin and diversification of major tetrapod lineages (sauropsids, synapsids and lissamphibians). This interval is represented by only a handful of tetrapod-bearing localities. Of these, the best understood is the classical tetrapod fauna of Joggins, Nova Scotia [1–4]. The Joggins fauna preserves a variety of early tetrapods, including an assortment of stem tetrapods and stem amphibians as well as the earliest members of the reptile (*Hylonomus lyelli*) and mammal (*Protoclepsydrops haplous*) lineages [4].

Despite the importance of the Joggins fauna, much of its diversity remains uncertain. Dawson assigned much of the diversity to the Microsauria [1–3], a higher taxon he established to accommodate several species each of *Hylonomus* and *Hylerpeton* [2]. Steen [5] updated much of this taxonomy, assigning many of Dawson's 'microsaurs' to the better accepted Lepospondyli [6] and establishing several new genera, but recognizing that many of Dawson's 'microsaurs' were not definitively assignable to any particular taxon. Carroll [7,8] further revised the taxonomy, acknowledging that many of the historic 'microsaurs' were likely not assignable to that group. Despite this history of work, many of the taxa established by Dawson (*Hylonomus*, *Hylerpeton*, *Dendrerpeton*) remain incompletely revised, probably obscuring the full diversity present at Joggins. Subsequent revisions of Joggins amniotes have suggested that at least some taxa attributed to early eureptiles (e.g. *Archerpeton*) may instead belong to 'microsaurs' [9] and vice versa [4], although recent phylogenetic studies have shown that the distinction between these two groups may be artificial [10]. However, many of the remaining taxa have not been revisited for some time, with taxonomic identifications reliant on obsolete phylogenetic frameworks. As a result, the ecological (upland versus lowland) and palaeobiogeographical context in which the earliest amniotes evolved remains largely uncertain.

While revising the Joggins 'microsaurs', we examined the holotype of *Hylerpeton longidentatum* Dawson 1876. This jaw was first reported by Dawson [2] as attributable to the 'microsaur' *Hylerpeton*, but distinguishable from the type species *Hylerpeton dawsoni* by the long, recurved dentition. Steen [5] and Carroll [8] both expressed doubts as to the microsaurian nature of the specimen, but both withheld judgement as to its taxonomic identity, although Carroll [8] noted similarities to non-tetrapod sarcopterygians. This specimen, which consists of the anterior half of a left lower jaw, is definitively not attributable to *Hylonomus*, 'microsaurs', or any temnospondyl, and instead represents the first aïstopod from Joggins, and from Canada more generally. Aïstopods are highly specialized limbless snake-like early tetrapods found abundantly in Late Pennsylvanian and Early Permian fossil localities across North America and Europe [11–13] and are one of the most abundant tetrapodomorph fossils found in some Carboniferous-aged sites, including Mazon Creek, IL, Linton, OH, and Five Points, OH. Aïstopods are present but rare in the Lower Carboniferous of Scotland [14,15], but recent assessment of their morphology and phylogenetics suggests this lineage had a Devonian origin among ichthyostegalian-grade stem tetrapods [10]. We here describe the Joggins aïstopod, finding surprisingly primitive morphology not previously reported in an aïstopod. This new fossil bridges the morphological gap between the fish-like stem-tetrapod outgroups and the snake-like Permo-Carboniferous aïstopods.

# 2. Material and methods

Specimens at the Redpath Museum in Montreal, Canada, were examined. In addition, the specimen RM 2.1129 was compared with aïstopod specimens from Mazon Creek, IL, and Linton, OH, including the genera *Oestocephalus* and *Phlegethontia*. We also compared the Joggins specimen with micro-computed tomography data of the jaws of the aïstopods *Lethiscus stocki* and *Coloraderpeton brilli* [10,13].

All specimens were photographed with a Nikon D700 camera using an AF-S NIKKOR 24–85 mm lens. Digital photographs were processed using Adobe Photoshop CS6. Figures were assembled using Adobe Illustrator CS6.

rsos.royalsocietypublishing.org    R. Soc. open sci. **5**: 181056

**Systematic Palaeontology**

Osteichthyes

Sarcopterygii

Tetrapodomorpha

Aïstopoda Miall 1875

*Andersonerpeton* gen. nov.

**LSID:** urn:lsid:zoobank.org:pub:1A86B126-3723-4821-ABC0-E382FEB1BA7A

**Etymology:** After Jason S. Anderson, whose work has modernized our understanding of aïstopod morphology and phylogeny. The Greek '-erpeton' (crawler) is a common name ending for early tetrapods.

**Diagnosis:** As for the type and only species

*Andersonerpeton longidentatum* comb. nov.

=*Hylerpeton longidentatum* Dawson 1876

**Hototype:** RM 2.1129, a nearly complete left mandible and associated dentition.

**Locality and Horizon:** Old Forest layer of the Joggins formation, Lyell's Corner, Joggins Fossil Cliffs, Joggins, Nova Scotia, Canada.

**Diagnosis**: Large aïstopod with the following combination of characters: coronoid denticle field; two anteriormost coronoids bear fang pairs; parasymphyseal plate with shagreen of denticles; prominent medial and lateral parasymphyseal foramina, singular prominent parasymphyseal tooth. Shares the presence of an adsymphyseal with *Coloraderpeton* and *Lethiscus*, but denticulate in contrast to the latter taxa. Shares a latero-medially wide dentary with *Coloraderpeton*, *Oestocephalus*, *Ophiderpeton* and *Lethiscus*.

**Comments:** Aside from this lower jaw, a number of other fossil tetrapod elements are present in the Joggins tree stump and are accessioned under the specimen number RM 2.1129. These include a number of fragmentary postcranial remains including ribs, an interclavicle, vertebrae and several isoated cranial bones. None of these appear attributable to an aïstopod or aïstopod-like tetrapod, however, and probably represent a small eureptile, such as *Hylonomus*, or a recumbirostran.

## 2.1. Description

The dentary is relatively narrow and thin in lateral aspect and tapers slightly anteriorly (figure 1). The external surface of the dentary is generally unornamented but shows a dense field of foramina ventral to the tooth row. A small row of the pustular ornament is present lateral to the tooth row along the anterior quarter of the jaw (figure 1*b*). The symphyseal region is dorsally recurved such that the occlusal surface of the symphyseal region faces somewhat posterodorsally rather than directly dorsally. The symphysis is composed primarily of the dentary, without an obvious contribution of the splenial on the lateral surface. A canal housing the Meckel's cartilage in the symphyseal region may be present. A prominent parasymphyseal fang is present lateral to the symphysis (figure 1*c*); the fang is approximately the same size as the marginal teeth. Marginal teeth are long and recurved without substantial heterodonty. Marginal teeth are all loosely connected to the underlying dentary, and many fold posteromedially, exaggerating the recurved appearance of the dentition. This does not appear to be taphonomic breakage, as the teeth all appear to be separated from the jaw at the same place, and no equivalent folding can be observed in the dentition of any other Joggins fossils. A similar weak articulation between the teeth and jaw can be seen in the aïstopod *C. brilli* [10,13] and possibly in an unnamed phlegethontiid from Montceau-les-Mines [16] and is observable in other oestocephalids [11], but has not to our knowledge been reported elsewhere in early tetrapods.

Just posterior to the parasymphyseal fang, an adsymphyseal sits in articulation with the dentary (figure 1*c*). The adsymphyseal is covered by a shagreen of denticles and extends posteriorly medial to the lateral parasymphyseal foramina to contact the anterior coronoid. The anterior coronoid is large and exhibits a prominent fang pair at its anteriormost extent. The coronoid fangs are approximately as large as the marginal teeth and are similarly folded posteriorly, suggesting a similar weak implantation zone. A broad denticle field posterior to the first coronoid represents the middle and posterior coronoid. A fang–pit pair is present on the anterior portion of the second coronoid, but no conspicuous fang–pit pair is present on the third coronoid. All three coronoids are separated from the tooth row by a trough of the dentary, which may have held the successional dental lamina and associated tissues. The anterior coronoid is bounded medially by a lamina of the splenial, whereas the middle and posterior coronoid are bounded medially by the prearticular. The splenial is perforated medially by a series of small foramina. These are unlikely to be a Meckelian foramen given the relatively anterodorsal position. Just ventral to these foramina is a small groove that delineates the dermal surface of the splenial from the visceral surface. This groove may represent a point of insertion for the intermandibularis musculature; the separation from the splenial foramina suggests that it does not represent the course of the efferent branchial arteries as preserved along

rsos.royalsocietypublishing.org    R. Soc. open sci. **5**: 181056

**Figure 1.** *Andersonerpeton longidentatum* comb. nov., RM 2.1129, left lower jaw. Scale bar equals 1 cm. (*a*) Ventral view, (*b*) left lateral view, (*c*) occlusal view, (*d*) medial view. Abbreviations: ?acden: postular ornament possibly equivalent to accessory denticles; ad: adsymphyseal; c1: first coronoid; c2: second coronoid; c3: third coronoid; cf: coronoid fang; d: dentary; dpsf: dentary parasymphyseal fang; lpsf: lateral parasymphyseal foramen; Mb: Meckelian bone; pa: prearticular; psp: postsplenial; spl: splenial.

the ventral margin of the lower jaw in some stem tetrapods. The postsplenial lacks a medial lamina and is mostly restricted to the ventral surface of the jaw. The anterior end of the prearticular is preserved medial to the second coronoid and appears to have contacted the splenial in life. A smooth rod-like element separates the prearticular from the infradentary bones ventrally, and probably represents a Meckelian bone.

rsos.royalsocietypublishing.org    R. Soc. open sci. **5**: 181056

# 3. Discussion

## 3.1. Position within early tetrapods

*Andersonerpeton longidentatum* shows several conspicuous aïstopod-like characteristics, particularly the weak, possibly ligamentous, articulation between the anterior dentary teeth and the underlying dentary bone. This is seen also in ophiderpetontids [11] and the specialized basal aïstopod *C. brilli* [10,13] and gives the appearance of a highly recurved anterior dentition. A similar weak contact between the marginal dentition and the underlying bone has also been reported in a phlegethontiid from Montceau-les-Mines [16]. Although parallels have been drawn between this weak implantation zone and lissamphibian pedicelly [16], no orthodentine pedicels are present, and instead mature teeth implant into alveoli in a manner comparable to that seen in snakes [17,18]. Interestingly, the marginal dentition of the elginerpetontids *Elginerpeton pancheni* and *Obruchevichthys gracilis* is typically not preserved, with most tooth loci represented only by open alveoli comparable in some ways to the aïstopod condition exemplified by *A. longidentatum* as well as previously described oestocephalids [19,20].

This is consistent with other features of *A. longidentatum*, which suggest not only a basal position within Aïstopoda but also a basal position within tetrapods more broadly. Conspicuously, teeth of *A. longidentatum* show labyrinthine infolding. Labyrinthine infolding is plesiomorphic for tetrapods but is absent in ophiderpetontids and *C. brilli* and is presumed to be absent in *L. stocki* and phlegethontiids. The presence of prominent parasymphyseal and coronoid fangs is unique and unexpected among aistopods as well but is consistent with the morphology seen in some stem tetrapods (figure 2), particularly *E. pancheni*, *Densignathus rowei*, *Metaxygnathus denticulatus*, *Ventastega curonica* and, to a lesser degree, *Ichthyostega* [32]. The presence of a robust Meckelian ossification exposed medially, dividing the prearticular from the postdentary bones, supports a close relationship with early stem tetrapods, suggesting that the aïstopod lineage diverged from crown tetrapods as early as the late Frasnian. Comparison with *C. brilli* [10,13] suggests a similar condition in that taxon, albeit with the Meckel's bone fused to the overlapping prearticular and, possibly, the articular. This compound bone may correspond to the 'posterior element' of phlegethontiids as well.

One specifically interesting character that should receive further attention is the fine pustular ornament just lateral to the dentary tooth row. Early tetrapodomorphs typically show a row of small denticles lateral to the dentary tooth row [21], but these denticles are lost early in the fin-to-limb transition, essentially appearing only in *E. pancheni* [19]. The ornament in *A. longidentatum* is similar in appearance, but it is unclear whether these pustules represent actual denticles, or simply a pustular ornament similar to that seen in an assortment of taxa spanning the fin-to-limb transition. If these pustules are in fact dental in origin, this would suggest an even earlier origin of aïstopods within the tetrapod stem, with aïstopods diverging from the tetrapod stem lineage prior to the divergence of ichthyostegalids, metaxygnathids and whatcheeriids. Given the absence of articulated autopodia in elginerpetontids, and the absence of differentiated digits in all taxa stemward of *Elginerpeton* [33], it may be worth considering whether the limbless aïstopods diverged from tetrapods prior to the origin of digits.

## 3.2. Aïstopods in the Joggins fauna

Aïstopods show a number of characteristics associated with an aquatic lifestyle such as well-developed cranial–lateral line canals and a spiracle [10], and are typically associated with aquatic-dominated faunas such as Mazon Creek, Linton, Nỳřany and Montceau-les-Mines [12,16]. However, they do sometimes occur in terrestrial-dominated faunas such as the Interval 300 Quarry [13] and Dolese Quarry [34,35] faunas, leading to uncertainty about the potential range of habitats occupied by aïstopods [12,16]. The presence of an aïstopod in the Old Forest layer at Joggins follows this pattern of the uncertain depositional environment. *A. longidentatum* comes from a bone-bearing deposit within an upright lycopsid stump along with other tetrapod remains. Although the stump deposits are thought to represent entirely terrestrial faunas, including terrestrial eureptiles [36], synapsids [4], 'microsaurs' [7] and temnospondyls [37], the stump-bearing horizons also contain aquatic embolomeres [38] as well as a range of chondrichthyans, dipnoans and actinopterygians [39], including fish and embolomere material within the stumps themselves (A. Mann and J.D. Pardo 2018, personal observation). Although the upright stump assemblages have been traditionally interpreted as evidence of refuge seeking by terrestrial animals during forest fires [40], the abundance of aquatic material suggests that this is a mixed attritional assemblage with both aquatic and terrestrial components, and that *A. longidentatum* could plausibly belong to either an aquatic or terrestrial component of the overall Joggins fauna.

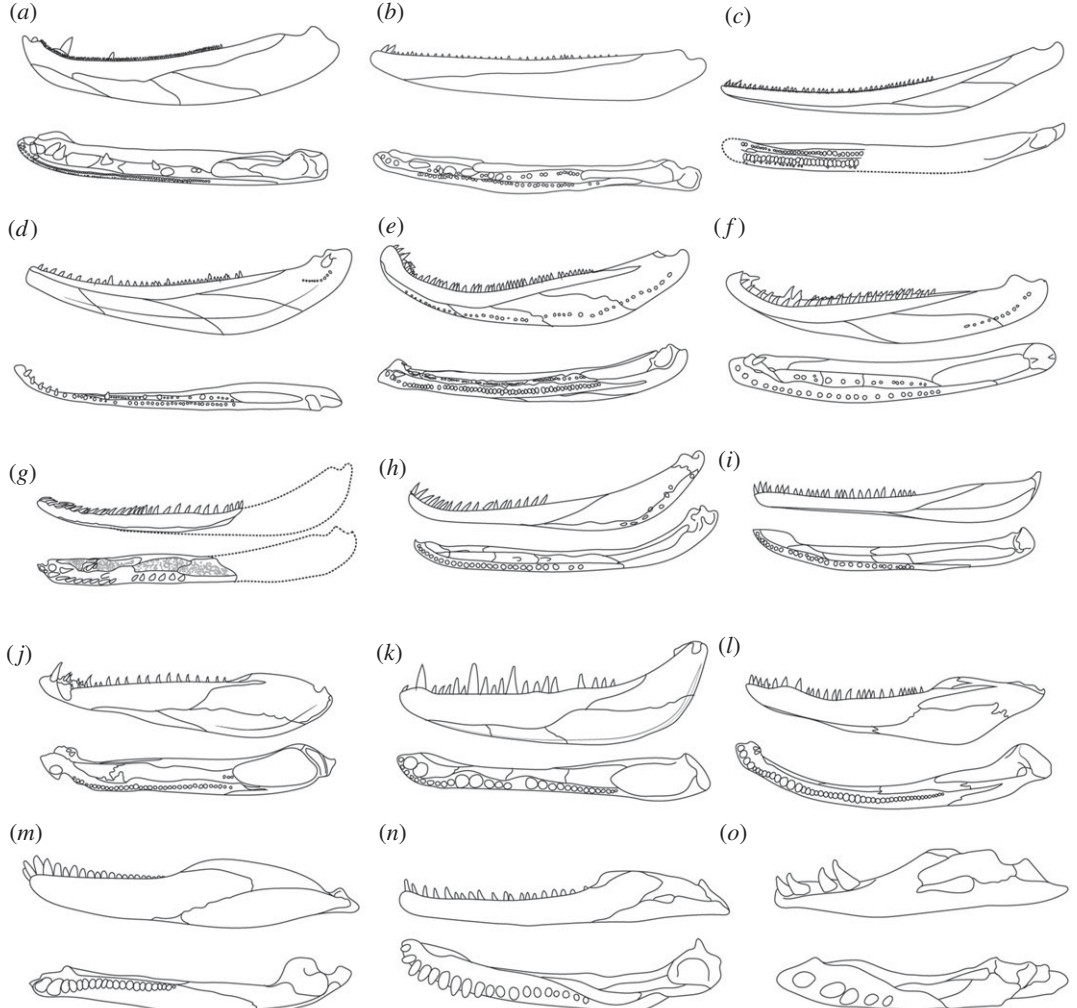

**Figure 2.** Comparative morphology of early tetrapod jaws in lateral (top), and occlusal (bottom) view. (*a*) *Eusthenopteron foordi*, after [21]; (*b*) *Tiktaalik roseae*, after [22]; (*c*) *Elginerpeton pancheni*, after [19,23]; (*d*) *Metaxygnathus denticulatus*, after [24]; (*e*) *Acanthostega gunnari*, after [25]; (*f*) *Densignathus rowei*, after [26]; (*g*) *Andersonerpeton longidentatum*, this study; (*h*) *Coloraderpeton brilli*, after [10]; (*i*) *Lethiscus stocki*, after [10]; (*j*) *Greererpeton burkemorani*, after [27]; (*k*) *Megalocephalus pachycephalus*, after [28]; (*l*) *Eryops megacephalus*, after [29]; (*m*) *Captorhinus laticeps*, after [30]; (*n*) *Huskerpeton englehorni*, after [31]; (*o*) *Brachydectes newberryi*, after [41]. Illustrations not to scale.

The presence of an aïstopod in the Joggins fauna brings Joggins more in line with faunal communities of the Upper Mississippian (e.g. East Kirkton) as well as typical faunas of the Carboniferous–Permian Transition. This is expected given the presence of two aïstopods (the plesiomorphic aïstopod *L. stocki* from Wardie Shale [14] and the ophiderpetontid *Ophiderpeton kirktonense* from East Kirkton [15]) in the Viséan of Scotland, and the diverse aïstopods of the Carboniferous–Permian Transition but fill a gap in the interval spanning the Serpukhovian and Bashkirian in which aïstopods and other small tetrapods are poorly represented. Although *A. longidentatum* is known only from a single specimen, this record nonetheless shows that previous workers have underestimated the taxonomic diversity of the Joggins fauna by placing most small-bodied vertebrates into unrevised wastebasket taxa. We expect that careful taxonomic revision of the Joggins material will continue to bring the Joggins fauna in line with classic vertebrate assemblages from both the Late Mississippian and the Carboniferous–Permian Transition.

Ethics. No ethics assessment was required prior to the completion of this research, as this study relied entirely on museum collections. Similarly, collecting permits were not required, as no field collections were made.

Data accessibility. All data, which include photographs and text description of fossils, are included in the paper.

Authors' contributions. J.D.P. and A.M. designed the study, collected and analysed the data, and wrote the paper.

Competing interests. All authors declare no competing interests.

Funding. No funding supported this research.

Acknowledgements. We thank H.C.E. Larsson and A. Howell (Redpath Museum, McGill University) for access to the specimen. We also thank two anonymous reviewers for providing helpful feedback.

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
