## [Reviewer comments · Royal Society Open Science]

Review History

RSOS-181056.R0 (Original submission)

Review form: Reviewer 1 (Jennifer Clack)

Is the manuscript scientifically sound in its present form?

Yes

Are the interpretations and conclusions justified by the results?

Yes

Is the language acceptable?

Yes

Is it clear how to access all supporting data?

Yes

Do you have any ethical concerns with this paper?

No

Have you any concerns about statistical analyses in this paper?

No

Recommendation?

Accept with minor revision (please list in comments)

Comments to the Author(s)

An interesting discovery. Figures of *Acanthostega* and *Densignathus* were switched in Fig. 2, also a couple of other minor inaccuracies. At least one results from choosing the scan over the drawing of the specimen for your data. A few upper and lower case letters need to be amended. Should not there be a ZooBank ID erected for your new genus?

Review form: Reviewer 2

Is the manuscript scientifically sound in its present form?

Yes

Are the interpretations and conclusions justified by the results?

Yes

Is the language acceptable?

Yes

Is it clear how to access all supporting data?

Yes

Do you have any ethical concerns with this paper?

No

Have you any concerns about statistical analyses in this paper?

No

Recommendation?

Accept with minor revision (please list in comments)

Comments to the Author(s)

This is a very interesting and valuable paper that carefully examines material from an iconic fossil assemblage - the Joggins tetrapod fauna - to unmask its true diversity, identifying the presence of an aistopod in the Joggins. This brings the Joggins into line with other faunal communities of the times. Additionally, the specimen exhibits numerous primitive features that place it at the base of Aistopoda and supports earlier work by the authors that aistopods are basal within Tetrapoda. A very nice piece of work.

There are a few minor typos throughout the manuscript; please see annotated PDF (Appendix A). Additionally, I would suggest a few other minor changes prior to publication:

1. There is virtually no description of the splenial and no mention of the postsplenial in the description, even though they are both figured. A brief description of both elements would be useful.

2. Is there any information available on the nature of the contacts (overlapping, interdigitated) between the bones?
3. The Meckelian ossification is discussed in the text but not labelled in Figure 1 - could this be added?
4. I think that e and f are reversed (in terms of the taxa that they are identified as) in Figure 2 - please check.

Congratulations on a nice paper!

Decision letter (RSOS-181056.R0)

03-Oct-2018

Dear Dr Pardo

On behalf of the Editors, I am pleased to inform you that your Manuscript RSOS-181056 entitled "A basal aïstopod from the earliest Pennsylvanian of Canada, and the antiquity of the first limbless tetrapod lineage" has been accepted for publication in Royal Society Open Science subject to minor revision in accordance with the referee suggestions. Please find the referees' comments at the end of this email.

The reviewers and handling editors have recommended publication, but also suggest some minor revisions to your manuscript. Therefore, I invite you to respond to the comments and revise your manuscript.

- Ethics statement

- Data accessibility

If you wish to submit your supporting data or code to Dryad (<http://datadryad.org/>), or modify your current submission to dryad, please use the following link:
<http://datadryad.org/submit?journalID=RSOS&manu=RSOS-181056>

- Competing interests

- Authors' contributions

- Acknowledgements

- Funding statement

Because the schedule for publication is very tight, it is a condition of publication that you submit the revised version of your manuscript before 12-Oct-2018. Please note that the revision deadline will expire at 00.00am on this date. If you do not think you will be able to meet this date please let me know immediately.

on behalf of Dr Robert Sansom (Associate Editor) and Prof. Jon Blundy (Subject Editor)
openscience@royalsociety.org

Associate Editor Comments to Author (Dr Robert Sansom):

Associate Editor: 1

Comments to the Author:

Thank to the authors for submission of this MS. Both authors were pleased with the quality of the work and have recommended very minor revisions. We look forward to receiving the revised manuscript.

Reviewer comments to Author:

Reviewer: 1

Comments to the Author(s)

An interesting discovery. Figures of *Acanthostega* and *Densignathus* were switched in Fig. 2, also a couple of other minor inaccuracies. At least one results from choosing the scan over the drawing of the specimen for your data. A few upper and lower case letters need to be amended. Should not there be a ZooBank ID erected for your new genus?

Reviewer: 2

Comments to the Author(s)

This is a very interesting and valuable paper that carefully examines material from an iconic fossil assemblage - the Joggins tetrapod fauna - to unmask its true diversity, identifying the presence of an aistopod in the Joggins. This brings the Joggins into line with other faunal communities of the times. Additionally, the specimen exhibits numerous primitive features that place it at the base of Aistopoda and supports earlier work by the authors that aistopods are basal within Tetrapoda. A very nice piece of work.

There are a few minor typos throughout the manuscript; please see annotated PDF. Additionally, I would suggest a few other minor changes prior to publication:

1. There is virtually no description of the splenial and no mention of the postsplenial in the description, even though they are both figured. A brief description of both elements would be useful.
2. Is there any information available on the nature of the contacts (overlapping, interdigitated) between the bones?
3. The Meckelian ossification is discussed in the text but not labelled in Figure 1 - could this be added?
4. I think that e and f are reversed (in terms of the taxa that they are identified as) in Figure 2 - please check.

Congratulations on a nice paper!

Author's Response to Decision Letter for (RSOS-181056.R0)

See Appendix B.

Decision letter (RSOS-181056.R1)

01-Nov-2018

Dear Dr Pardo,

I am pleased to inform you that your manuscript entitled "A basal aistopod from the earliest Pennsylvanian of Canada, and the antiquity of the first limbless tetrapod lineage" is now accepted for publication in Royal Society Open Science.

on behalf of Dr Robert Sansom (Associate Editor) and Prof. Jon Blundy (Subject Editor)
openscience@royalsociety.org

Appendix A**ROYAL SOCIETY
OPEN SCIENCE****A basal aïstopod from the earliest Pennsylvanian of Canada,
and the antiquity of the first limbless tetrapod lineage**

Journal:	Royal Society Open Science
Manuscript ID	RSOS-181056
Article Type:	Research
Date Submitted by the Author:	13-Jul-2018
Complete List of Authors:	Pardo, Jason; University of Calgary, Department of Comparative Biology and Experimental Medicine Mann, Arjan; Carleton University, Earth Sciences
Subject:	Palaeontology < EARTH SCIENCES
Keywords:	Tetrapodamorphia, Carboniferous, Aïstopoda, Joggins, Lepospondyli
Subject Category:	Earth science

1 **A basal aïstopod from the earliest Pennsylvanian of Canada, and the antiquity of the first limbless**
2 **tetrapod lineage**

4 Jason D. Pardo^{1,2} and Arjan Mann³
5

6 ¹Department of Comparative Biology & Experimental Medicine, University of Calgary, Calgary, Canada.

[revised manuscript text omitted]

**Author contributions**

JDP and AM designed the study, JDP and AM collected and analyzed the data, JDP and AM wrote the
paper.

**Ethics**

No ethics assessment was required prior to the completion of this research, as this study relied entirely on
museum collections. Similarly, collecting permits were not required, as no field collections were made.

**Data availability**

All data, which include photographs and text description of fossils, are included in the paper.

**Competing interests**

All authors declare no competing interests.

**Funding**

No funding supported this research.

**Acknowledgments**

We thank H. C. E. Larsson and A. Howell (Redpath Museum, McGill University) for access to the
specimen.

**References**

1. Dawson JW. 1863. Air-breathers of the coal period: a descriptive account of the remains of land
animals found in the coal formation of Nova Scotia, with remarks on their bearing on theories of
the formation of coal and of the origin of species. Montreal, Dawson Brothers.

2. Dawson JW. 1876. On a recent discovery of Carboniferous batrachians in Nova Scotia. *American*
*Journal of Science* **12**, 440-447.

3. Dawson JW. 1882. The results of recent explorations of erect trees containing animal remains in
the Coal formation of Nova Scotia. *Philosophical Transactions of the Royal Society* **173**, 621-
659.

4. Carroll R. L. (1964). The earliest reptiles. *Zool. J. Linn Soc.* 45: 61-83.

5. Steen MC. 1934. The amphibian fauna from the South Joggins, Nova Scotia. *Proceedings of the*
*Zoological Society of London* **104**, 465-504.

6. Zittel K. 1888. *Handbuch der Paläontologie. Abteilung 1. Paläontologie Band III. Vertebrata*
(*Piscesm Amphibia, Reptilia, Aves*). Oldenbourg, München and Leipzig, 900 pg.
7. Carroll RL. 1963. A microsaur from the Pennsylvanian of Joggins, Nova Scotia. *Papers of the*
*National Museum of Canada* **22**, 1-13.
8. Carroll R. 1966. Microsaur from the Westphalian B of Joggins, Nova Scotia. *Proceedings*
*Linnean Society London* **177**, 63-97.
9. Reisz RR, Modesto SP. 1996. *Archerpeton anthracos* from the Joggins Formation of Nova
Scotia: a microsaur, not a reptile. *Canadian Journal of Earth Sciences* **33**, 703-709.
10. Pardo JD, Szostakiwskyj M, Ahlberg PE, Anderson JS. (2017) Hidden morphological diversity
among early tetrapods. *Nature* **546**, 642-645.
11. Carroll RL. 1998. Cranial anatomy of ophiderpetontid aïstopods: Palaeozoic limbless amphibians.
*Zoological Journal of the Linnean Society* **122**, 143-166.
12. Anderson JS. 2002. Revision of the aïstopod genus *Phlegethontia* (Tetrapoda: Lepospondyli).
*Journal of Paleontology* **76**, 1029-1046.
13. Anderson JS, Carroll RL, Rowe TB. 2003. New information on *Lethiscus stocki* (Tetrapoda:
Lepospondyli: Aistopoda) from high-resolution computed tomography and a phylogenetic
analysis of Aistopoda. *Canadian Journal of Earth Sciences* **40**, 1071-1083.
14. Wellstead CF. 1982. A Lower Carboniferous aïstopod amphibian from Scotland. *Palaeontology*
**25**, 193-208.
15. Milner AC. 1993. The aïstopod amphibian from the Viséan of East Kirkton, West Lothian,
Scotland. *Earth and Environmental Science Transactions of The Royal Society of Edinburgh* **84**,
363-368.
16. Germain D. 2008. A new phlegethontiid specimen (Lepospondyli, Aistopoda) from the Late
Carboniferous of Montceau-les-Mines (Saône-et-Loire, France). *Geodiversitas* **30**, 669-680.
17. Zaher H, Rieppel O. 1999. Tooth implantation and replacement in squamates, with special
reference to mosasaur lizards and snakes. *American Museum Novitates* **3721**, 1-19.

18. Savitsky AH. 1981. Hinged teeth in snakes: an adaptation for swallowing hard-bodied prey.
*Science* **212**,346-349.
19. Ahlberg PE. 1995. Elginerpeton pancheni and the earliest tetrapod clade. *Nature* **373**, 420-425.
20. Clément G, Lebedev O. 2014. Revision of the early tetrapod *Obruchevichthys* Vorobyeva, 1977
from the Frasnian (Upper Devonian) of the North-western East European Platform.
*Paleontological Journal* **48**, 1082-1091.
21. Ahlberg PE, Clack JA. 1998. Lower jaws, lower tetrapods-a review based on the Devonian genus
*Acanthostega*. *Transactions of the Royal Society of Edinburgh Earth and Environmental Science*
**89**, 11-46.
22. Porro LB, Rayfield EJ, Clack JA. 2015. Computed tomography, anatomical description and three-
dimensional reconstruction of the lower jaw of *Eusthenopteron foordi* Whiteaves, 1881 from the
Upper Devonian of Canada. *Palaeontology* **58**, 1031-1047.
23. Shubin NH, Daeschler EB, Jenkins FA Jr. 2006. The pectoral fin of *Tiktaalik roseae* and the
origin of the tetrapod limb. *Nature* **440**, 764-771.
24. McGinnis HJ. 1967. The osteology of *Phlegethontia*, a Carboniferous and Permian aïstopod
amphibian. *University of California Publications in Geological Sciences* **71**, 1-46.
25. Lund R. 1978. Anatomy and relationships of the family Phlegethontiidae (Amphibia, Aïstopoda).
*Annals of the Carnegie Museum* **47**, 53-79.
26. Carroll RL, Baird D. 1972. Carboniferous stem-reptiles of the family Romeriidae. *Bulletin of the*
*Museum of Comparative Zoology* **143**, 321-364.
27. Carroll RL. 1967. Labyrinthodonts from the Joggins Formation. *Journal of Paleontology* **41**, 111-
142.
28. Holmes RB, Carroll RL. 2010. An articulated embolomere skeleton (Amphibia: Anthracosauria)
from the Lower Pennsylvanian (Bashkirian) of Nova Scotia. *Canadian Journal of Earth Sciences*
**47**, 209-219.

29. Carpenter DK., Falcon-Lang HJ, Benton MJ, Grey M. 2015. Early Pennsylvanian (Langsettian)
fish assemblages from the Joggins Formation, Canada, and their implications for palaeoecology
and palaeogeography. *Palaeontology* **58**, 661-690.
30. Falcon-Lang HJ. 1999. Fire ecology of a late Carboniferous floodplain, Joggins, Nova Scotia.
*Journal of the Geological Society* 156, 137-148.
31. Daeschler EB, Shubin NH, Jenkins FA Jr. 2006. A Devonian tetrapod-like fish and the evolution of
the tetrapod body plan. *Nature* 440, 757-763.
32. Ahlberg PE, Friedman M, Blom H. 2005. New light on the earliest known tetrapod jaw. *Journal of*
*Vertebrate Paleontology* 25
33. Campbell KSW, Bell MW. 1977. A primitive amphibian from the Late Devonian of New South
Wales. *Alcheringa: An Australasian Journal of Palaeontology* 1, 369-381.
34. Daeschler EB. 2000. Early tetrapod jaws from the Late Devonian of Pennsylvania, USA. *Journal of*
*Paleontology* 74, 301-308.
35. Porro LB, Rayfield EJ, Clack JA. 2015. Descriptive anatomy and three-dimensional reconstruction
of the skull of the early tetrapod *Acanthostega gunnari* Jarvik, 1952. *PLOS ONE* 10, e0124731.
36. Bolt JR, Lombard RE. 2001. The mandible of the primitive tetrapod *Greererpeton*, and the early
evolution of the tetrapod lower jaw. *Journal of Paleontology* 75, 1016-1042.
37. Beaumont EH. 1977. Cranial morphology of the Loxommatidae (Amphibia: Labyrinthodontia).
*Philosophical Transactions of the Royal Society B* 280, 29-101.
38. Sawin HJ. 1941. The cranial anatomy of *Eryops megacephalus*. *Bulletin of the Museum of*
*Comparative Zoology* 88, 405-463.
39. Heaton MJ. 1979. Cranial anatomy of primitive captorhinid reptiles from the Late Pennsylvanian
and Early Permian Oklahoma and Texas. *Oklahoma Geological Survey Bulletin* 127, 1-84.

- 40. Huttenlocker AK, Pardo JD, Small BJ, Anderson JS. 2013. Cranial morphology of recumbirostrans
 (Lepospondyli) from the Permian of Kansas and Nebraska, and early morphological evolution
 inferred by micro-computed tomography. *Journal of Vertebrate Paleontology* 33, 540-552.
- 41. Pardo JD, Anderson JS. 2016. Cranial morphology of the Carboniferous-Permian tetrapod
 *Brachydectes newberryi* (Lepospondyli, Lysorophia): new data from μ CT. *PLOS ONE* 11,
 e0161823.

**Figures**

Figure 1. *Andersonerpeton longidentatum* comb. nov., RM 2.1129, left lower jaw. Scale bar equals 1 cm.
 (a) ventral view, (b) left lateral view, (c) occlusal view, (d) medial view. Abbreviations: ?acden; postular
 ornament possibly equivalent to accessory denticles; ad, adsymphyseal; c1, first coronoid; c2, second
 coronoid; c3, third coronoid; cf, coronoid fang; d, dentary; dpsf, dentary parasymphyseal fang; lpsf,
 lateral parasymphyseal foramen; psp, postsplenial; spl, splenial

Figure 2. Comparative morphology of early tetrapod jaws in lateral (top), and occlusal (bottom) view. a,
 *Eusthenopteron foordi*, after [22]; b, *Tiktaalik roseae*, after [31]; c, *Elginerpeton pancheni*, after [19,32];
 352 d, *Metaxygnathus denticulatus*, after [33]; e, *Densignathus rowei*, after [34]; f, *Acanthostega gunnari*,
 after [35]; g, *Andersonerpeton longidentatum*, this study; h, *Coloraderpeton brilli*, after [10]; i, *Lethiscus*
 *stocki*, after [10]; j, *Greererpeton burkemorani*, after [36]; k, *Megalocephalus pachycephalus*, after [37];
 355 l, *Eryops megacephalus*, after [38]; m, *Captorhinus laticeps*, after [39]; n, *Huskerpeton englehorni*, after
 356 [40]; o, *Brachydectes newberryi*, after [41]. Illustrations not to scale.

Figure 1. *Andersonerpeton longidentatum* comb. nov., RM 2.1129, left lower jaw. Scale bar equals 1 cm. (a)
ventral view, (b) left lateral view, (c) occlusal view, (d) medial view. Abbreviations: ?acden; postular
ornament possibly equivalent to accessory denticles; ad, adsymphyseal; c1, first coronoid; c2, second
coronoid; c3, third coronoid; cf, coronoid fang; d, dentary; dpsf, dentary parasymphyseal fang; lpsf, lateral
parasymphyseal foramen; psp, postsplenial; spl, splenial.

178x279mm (300 x 300 DPI)

Figure 2. Comparative morphology of early tetrapod jaws in lateral (top), and occlusal (bottom) view. a, *Eusthenopteron foordi*, after [22]; b, *Tiktaalik roseae*, after [31]; c, *Elginerpeton pancheni*, after [19,32]; d, *Metaxygnathus denticulatus*, after [33]; e, *Densignathus rowei*, after [34]; f, *Acanthostega gunnari*, after [35]; g, *Andersonerpeton longidentatum*, this study; h, *Coloraderpeton brilli*, after [10]; i, *Lethiscus stocki*, after [10]; j, *Greererpeton burkemorani*, after [36]; k, *Megalocephalus pachycephalus*, after [37]; l, *Eryops megacephalus*, after [38]; m, *Captorhinus laticeps*, after [39]; n, *Huskerpeton englehorni*, after [40]; o, *Brachydictes newberryi*, after [41]. Illustrations not to scale.

151x144mm (300 x 300 DPI)

Appendix B

Dear Editor,

Please see our revised manuscript. We have taken care to address all reviewer comments. In particular, we have revised Figure 1 to more clearly figure the Meckelian bone (unclear in the original drawings) and have revised Figure 2 to correct taxonomic confusion and to address some minor anatomical differences that one of the reviewers noted. We have also included a zoobank ID to associate with the nomenclatural act made in this manuscript.